# Powderworld: A Platform for Understanding Generalization via Rich Task Distributions

**Kevin Frans**
MIT CSAIL
`kvfrans@mit.edu`

**Phillip Isola**
MIT CSAIL
`phillipi@mit.edu`

## Abstract

One of the grand challenges of reinforcement learning is the ability to generalize to new tasks. However, general agents require a set of rich, diverse tasks to train on. Designing a 'foundation environment' for such tasks is tricky – the ideal environment would support a range of emergent phenomena, an expressive task space, and fast runtime. To take a step towards addressing this research bottleneck, this work presents Powderworld, a lightweight yet expressive simulation environment running directly on the GPU. Within Powderworld, two motivating challenges distributions are presented, one for world-modelling and one for reinforcement learning. Each contains hand-designed test tasks to examine generalization. Experiments indicate that increasing the environment's complexity improves generalization for world models and certain reinforcement learning agents, yet may inhibit learning in high-variance environments. Powderworld aims to support the study of generalization by providing a source of diverse tasks arising from the same core rules. Try an interactable demo at kvfrans.com/static/powder

## 1 Introduction

One of the grand challenges of reinforcement learning (RL), and of decision-making in general, is the ability to generalize to new tasks. RL agents have shown incredible performance on single task settings (Berner et al., 2019; Lillicrap et al., 2015; Mnih et al., 2013), yet frequently stumble when presented with unseen challenges. Single-task RL agents are largely overfit on the tasks they are trained on (Kirk et al., 2021), limiting their practical use. In contrast, a general agent, which can robustly perform well on a wide range of novel tasks, can then be adapted to solve downstream tasks and unseen challenges.

General agents greatly depend on a diverse set of tasks to train on. Recent progress in deep learning has shown that as the amount of data increases, so do generalization capabilities of trained models (Brown et al., 2020; Ramesh et al., 2021; Bommasani et al., 2021; Radford et al., 2021). Agents trained on environments with domain randomization or procedural generation capabilities transfer better to unseen test tasks Cobbe et al. (2020); Tobin et al. (2017); Risi & Togelius (2020); Khalifa et al. (2020). However, as creating training tasks is expensive and challenging, most standard environments are inherently over-specific or limited by their focus on a single task type, e.g. robotic control or gridworld movement.

As the need to study the relationships between training tasks and generalization increases, the RL community would benefit greatly from a 'foundation environment' supporting diverse tasks arising from the same core rules. The benefits of expansive task spaces have been showcased in Unsupervised Environment Design (Wang et al., 2019; Dennis et al., 2020; Jiang et al., 2021; Parker-Holder et al., 2022), but gridworld domains fail to display how such methods scale up. Previous works have proposed specialized task distributions for multi-task training (Samvelyan et al., 2021; Suarez et al., 2019; Fan et al., 2022; Team et al., 2021), each focusing on a specific decision-making problem. To further investigate generalization, it is beneficial to have an environment where many variations of training tasks can easily be compared.

As a step toward lightweight yet expressive environments, this paper presents Powderworld, a simulation environment geared to support procedural data generation, agent learning, and multi-task generalization. Powderworld aims to efficiently provide environment dynamics by running directly

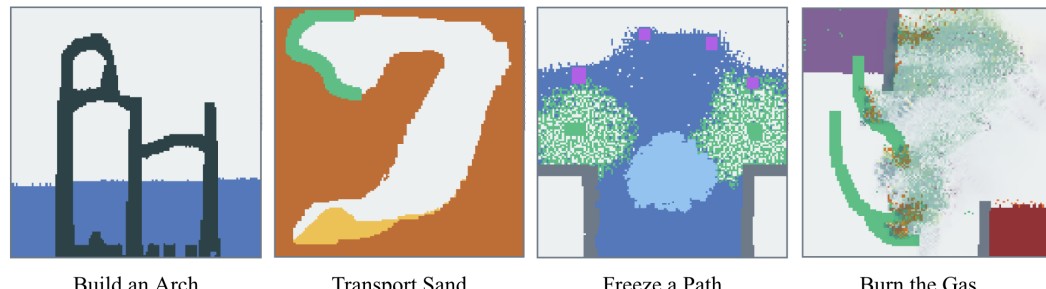

| Build an Arch | Transport Sand | Freeze a Path | Burn the Gas |

Figure 1: **Examples of tasks created in the Powderworld engine.** Powderworld provides a physics-inspired simulation over which many distributions of tasks can be defined. Pictured above are human-designed challenges where a player must construct unstable arches, transport sand through a tunnel, freeze water to create a bridge, and draw a path with plants. Tasks in Powderworld creates challenges from a set of core rules, allowing agents to learn generalizable knowledge. **Try an interactive Powderworld simulation at** kvfrans.com/static/powder

on the GPU. Elements (e.g. sand, water, fire) interact in a modular manner within local neighborhoods, allowing for efficient runtime. The free-form nature of Powderworld enables construction of tasks ranging from simple manipulation objectives to complex multi-step goals. Powderworld aims to 1) be modular and supportive of emergent interactions, 2) allow for expressive design capability, and 3) support efficient runtime and representations.

Additionally presented are two motivating frameworks for defining world-modelling and reinforcement learning tasks within Powderworld. World models trained on increasingly complex environments show superior transfer performance. In addition, models trained over more element types show stronger fine-tuning on novel rulesets, demonstrating that a robust representation has been learned. In the reinforcement learning case, increases in task complexity benefit generalization up to a task-specific inflection point, at which performance decreases. This point may mark when variance in the resulting reward signal becomes too high, inhibiting learning. These findings provide a starting point for future directions in studying generalization using Powderworld as a foundation.

## 2    RELATED WORK

**Task Distributions for RL.** Video games are a popular setting for studying multi-task RL, and environments have been built off NetHack (Samvelyan et al., 2021; Küttler et al., 2020), Minecraft (Fan et al., 2022; Johnson et al., 2016; Guss et al., 2019), Doom (Kempka et al., 2016), and Atari (Bellemare et al., 2013). Team et al. (2021); Yu et al. (2020); Cobbe et al. (2020) describe task distributions focused on meta-learning, and Fan et al. (2022); Suarez et al. (2019); Hafner (2021); Perez-Liebana et al. (2016) detail more open-ended environments containing multiple task types. Most similar to this work may be ProcGen (Cobbe et al., 2020), a platform that supports infinite procedurally generated environments. However, while ProcGen games each have their own rulesets, Powderworld aims to share core rules across all tasks. Powderworld focuses specifically on runtime and expressivity, taking inspiration from online "powder games" where players build ranges of creations out of simple elements (bal; pow; Bittker).

**Generalization in RL.** Multi-task reinforcement learning agents are generally valued for their ability to perform on unseen training tasks (Packer et al., 2018; Kirk et al., 2021). The sim2real problem requires agents aim to generalize to out-of-distribution real world domains (Tobin et al., 2017; Sadeghi & Levine, 2016). The platforms cited above also target generalization, often within the context of solving unseen levels within a game. This work aims to study generalization within a physics-inspired simulated setting, and creates out-of-distribution challenges by hand-designing a set of unseen test tasks.

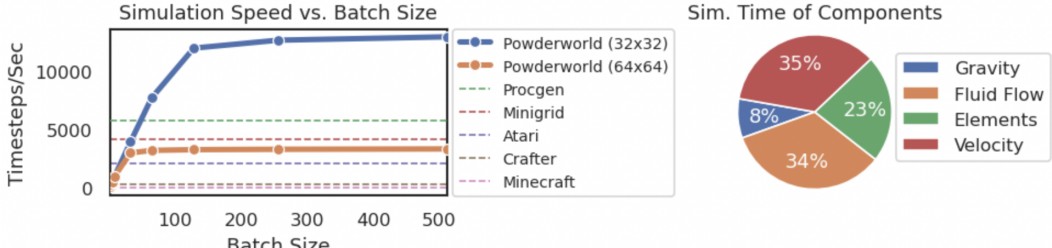

Figure 2: **Powderworld runs on the GPU and can simulate many worlds in parallel.** GPU simulation provides a significant speedup and allows simulation time to scale with batch size. Simulation speed is guaranteed to remain constant regardless of how many elements are present in the world.

## 3    POWDERWORLD ENVIRONMENT

The main contribution of this work is an environment specifically for training generalizable agents over easily customizable distributions of tasks. Powderworld is designed to feature:

- **Modularity and support for emergent phenomena.** The core of Powderworld is a set of fundamental rules defining how two neighboring elements interact. The consistent nature of these rules is key to agent generalization; e.g. fire will always burn wood, and agents can learn these inherent properties of the environment. Furthermore, local interactions can build up to form emergent wider-scale phenomena, e.g. fire spreading throughout the world. This capacity for emergence enables tasks to be diverse yet share consistent properties. Thus, fundamental Powderworld priors exist that agents can take advantage of to generalize.

- **Expressive task design capability.** A major challenge in the study of RL generalization is that tasks are often nonadjustable. Instead, an ideal environment should present an explorable space of tasks, capable of representing interesting challenges, goals, and constraints. Tasks should be parametrized to allows for automated design and interpretable control. Powderworld represents each task as a 2D array of elements, enabling a variety of procedural generation methods. Many ways exist to test a specific agent capability, e.g. "burn plants to create a gap", increasing the chance that agents encounter these challenges.

- **Fast runtime and representation.** As multi-task learning can be computationally expensive, it is important that the underlying environment runs efficiently. Powerworld is designed to run on the GPU, enabling large batches of simulation to be run in parallel. Additionally, Powderworld employs a neural-network-friendly matrix representation for both task design and agent observations. To simplify the training of decision-making agents, the Powderworld representation is fully-observable and runs on a discrete timescale (but partial-observability is an easy modification if desired).

### 3.1    ENGINE

Described below is an overview of the engine used for the Powderworld simulator. Additional technical details can be founded in the Appendix.

**World matrix.** The core structure of Powderworld is a matrix of elements $W$ representing the world. Each location $W_{x,y}$ holds a vector of information representing that location in the world. Namely, each vector contains a one-hot encoding of the occupying element, plus additional values indicating gravity, density, and velocity. The $W$ matrix is a Markovian state of the world, and thus past $W$ matrices are unnecessary for state transitions. Every timestep, a new $W$ matrix is generated via a stochastic update function, as described below.

**Gravity.** Certain elements are affected by gravity, as noted by the IsGravity flag in Figure 3. Each gravity-affected element also holds a density value, which determines the element's priority during the gravity calculation. Every timestep, each element checks with its neighbor below. If both elements are gravity-affected, and the neighbor below has a lower density, then the two elements swap positions. This interaction functions as a core rule in the Powderworld simulation and allows elements to stack, displace, and block each other.

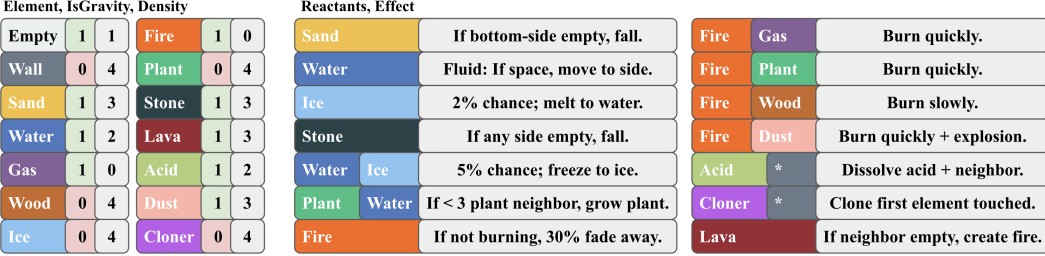

**Element, IsGravity, Density**

| Element | IsGravity | Density | Element | IsGravity | Density |
|---|---|---|---|---|---|
| Empty | 1 | 1 | Fire | 1 | 0 |
| Wall | 0 | 4 | Plant | 0 | 4 |
| Sand | 1 | 3 | Stone | 1 | 3 |
| Water | 1 | 2 | Lava | 1 | 3 |
| Gas | 1 | 0 | Acid | 1 | 2 |
| Wood | 0 | 4 | Dust | 1 | 3 |
| Ice | 0 | 4 | Cloner | 0 | 4 |

**Reactants, Effect**

| Reactants | | Effect |
|---|---|---|
| Sand | | If bottom-side empty, fall. |
| Water | | Fluid: If space, move to side. |
| Ice | | 2% chance; melt to water. |
| Stone | | If any side empty, fall. |
| Water | Ice | 5% chance; freeze to ice. |
| Plant | Water | If < 3 plant neighbor, grow plant. |
| Fire | | If not burning, 30% fade away. |

| Reactants | | Effect |
|---|---|---|
| Fire | Gas | Burn quickly. |
| Fire | Plant | Burn quickly. |
| Fire | Wood | Burn slowly. |
| Fire | Dust | Burn quickly + explosion. |
| Acid | * | Dissolve acid + neighbor. |
| Cloner | * | Clone first element touched. |
| Lava | | If neighbor empty, create fire. |

Figure 3: **A list of elements and reactions in the Powderworld simulation.** Elements each contain gravity and density information. A set of element-specific reactions dictates how each element behaves and reacts to neighbors. Certain reactions manipulate the world's velocity field, which can push further elements away. Together, the gravity, velocity, and reaction systems create a core set of rules by which interesting simulations arise.

**Element-specific reactions.** The behavior of Powderworld arises from a set of modular, local element reactions. Element reactions can occur either within a single element, or as a reaction when two elements are neighbors to each other. These reactions are designed to facilitate larger-scale behaviors; e.g. the sand element falls to neighboring locations, thus areas of sand form pyramid-like structures. Elements such as water, gas, and lava are fluids, and move horizontally to occupy available space. Finally, pairwise reactions provide interactions between specific elements, e.g. fire spreads to flammable elements, and plants grow when water is nearby. See Figure 3 for a description of the Powderworld reactions, and full documentation is given in the appendix and code.

**Velocity system.** Another interaction method is applying movement through the velocity system. Certain reactions, such as fire burning or dust exploding, add to the velocity field. Velocity is represented via an two-component $V_{x,y}$ vector at each world location. If the magnitude of the velocity field at a location is greater than a threshold, elements are moved in one of eight cardinal directions, depending on the velocity angle. Velocity naturally diffuses and spreads in its own direction, thus a velocity difference will spread outwards before fading away. Walls are immune to velocity affects. Additionally, the velocity field can be directly manipulated by an interacting agent.

All operators are local and translation equivariant, yielding a simple implementation in terms of (nonlinear) convolutional kernels. To exploit GPU-optimized operators, Powderworld is implemented in Pytorch (Paszke et al., 2019), and performance scales with GPU capacity (Figure 2).

## 4 EXPERIMENTS

The following section presents a series of motivating experiments showcasing task distributions within Powderworld. These tasks intend to provide two frameworks for accessing the richness of the Powderworld simulation, one through supervised learning and one through reinforcement learning. While these tasks aim to specifically highlight how Powderworld can be used to generate diverse task distributions, the presented tasks are by no means exhaustive, and future work may easily define modifications or additional task objectives as needed.

In all tasks, the model is provided the $W \in \mathbb{R}^{H \times W \times 20}$ matrix as an observation, which is a Markovian state containing element, gravity, density, and velocity information. All task distributions also include a procedural generation algorithm for generating training tasks, as well as tests used to measure transfer learning.

*In all experiments below, evaluation is on out-of-distribution tests which are unseen during training.*

### 4.1 WORLD MODELLING TASK

This section examines a world-modelling objective in which a neural network is given an observation of the world, and must then predict a future observation. World models can be seen as learning how to encode an environment's dynamics, and have proven to hold great practical value in downstream decision making (Ha & Schmidhuber, 2018; Hafner et al., 2019b;a). A model which can correctly predict the future of any observation can be seen as thoroughly understanding the core rules of

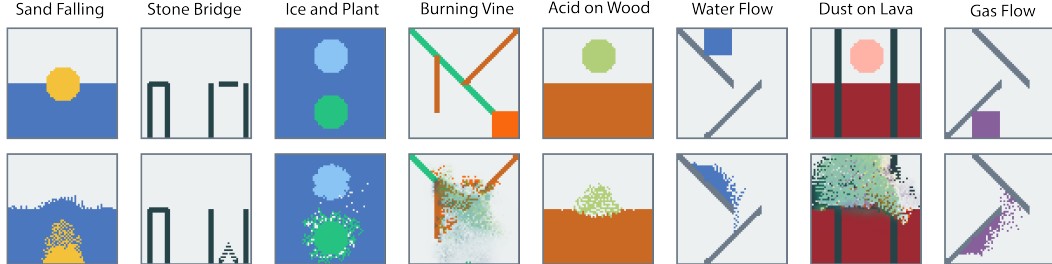

Figure 4: **World modelling test states are designed to showcase specific element interactions.** Test states are out-of-distribution and unseen during training. Model generalization capability is measured by how accurate its future predictions are on all eight tests.

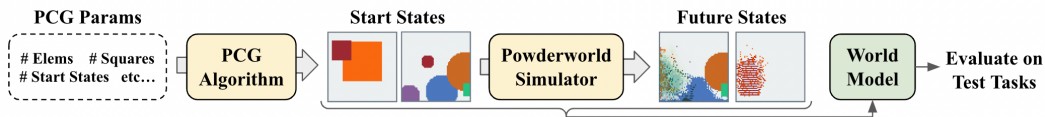

Figure 5: **Training states are generated via a procedural content generation (PCG) algorithm followed by Powderworld simulation.** Experiments examine the affect of increasing complexity in PCG parameters.

the environment. The world-modelling task does not require reinforcement learning, and is instead solved via a supervised objective with the future state as the target.

Specifically, given an observation $W^0 \in \mathbb{R}^{H \times W \times N}$ of the world, the model is tasked with generating a $W' \in \mathbb{R}^{H \times W \times N}$ matrix of the world 8 timesteps in the future. $W'$ values corresponds to logit probabilities of the $N$ different elements, and loss is computed via cross-entropy between the true and predicted world. Tasks are represented by a tuple of starting and ending observations.

Training examples for the world-modelling task are created via an parametrized procedural content generation (PCG) algorithm. The algorithm synthesizes starting states by randomly selecting elements and drawing a series of lines, circles, and squares. Thus, the training distribution can be modified by specifying how many of each shape to draw, out of which elements, and how many total starting states should be generated. A set of hand-designed tests are provided as shown in Figure 4 which each measures a distinct property of Powderworld, e.g. simulate sand falling through water, fire burning a vine, or gas flowing upwards. To generate the targets, each starting state is simulated forwards for 8 timesteps, as shown in Figure 5.

The model is a convolutional U-net network (Ronneberger et al., 2015), operating over a world size of 64x64 and 14 distinct elements. The agent network consists of three U-net blocks with 32, 64, and 128 features respectively. Each U-net block contains two convolutional kernels with a kernel size of three and ReLU activation, along with a MaxPool layer in the encoder blocks. The model is trained with Adam for 5000 iterations with a batch size of 256 and learning rate of 0.005. During training, a replay buffer of 1024*256 data points is randomly sampled to form the training batch, and the oldest data points are rotated out for fresh examples generated via the Powderworld simulator.

### 4.1.1 CAN WORLD MODELS GENERALIZE TO UNSEEN TEST STATES?

A starting experiment examines whether world models trained purely on simulated data can correctly generalize on hand-designed test states. The set of tests, as shown in Figure 4, are out-of-distribution hand-designed worlds that do not appear in the training set. A world model must discover the core ruleset of environmental dynamics in order to successfully generalize.

Scaling laws for training large neural networks have shown that more data consistently improves performance (Kaplan et al., 2020; Zhai et al., 2022). Figure 6 shows this observation to be true in Powderworld as well; world models trained on increasing amounts of start states display higher performance on test states. Each world model is trained on the same number of training examples

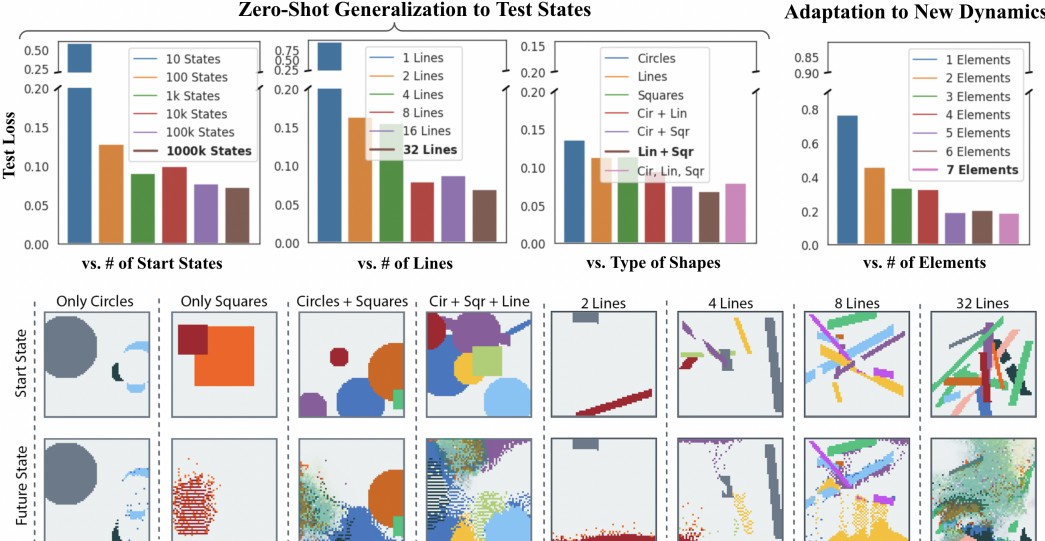

Figure 6: **World model generalization improves as training distribution complexity is increased.** Shown are the test performances of world models trained with data from varying numbers of start states, number of lines, and types of shapes. By learning from diverse data, world models can better generalize to unseen test states. **Top-Right: World models trained on more elements can better fine-tune to novel elements.** These results show that Powderworld provides a rich enough simulation that world models learn robust representations capable of adaptation to new dynamics. **Bottom:** Examples of states generated with various PCG parameters.

and timesteps, the only difference is how this data is generated. The average test loss over three training runs are displayed.

Results show that the 10-state world model overfits and does not generalize to the test states. In contrast, the 100-state model achieves much higher test accuracy, and the trend continues as the number of training tasks improves. These results show that the Powderworld world-modelling task demonstrates similar scaling laws as real-world data.

### 4.1.2 HOW DO INCREASINGLY COMPLEX TRAINING TASKS AFFECT GENERALIZATION?

As training data expands to include more varieties of starting states, does world model performance over a set of test states improve? More complex training data may allow world models to learn more robust representations, but may also introduce variance which harms learning or create degenerate training examples when many elements overlap.

Figure 6 displays how as additional shapes are included within the training distribution, zero-shot test performance successfully increases. World models are trained on distributions of training states characterized by which shapes are present between lines, circles, and square. Lines are assigned a random $(X^1,Y^1)$, $(X^2,Y^2)$, and thickness. Circles and Squares are assigned a random $(X^1,Y^1)$ along with a radius. Each shape is filled in with a randomly selected element. Between 0 and 5 of each shape are drawn. Interestingly, training tasks with less shape variation also display higher instability, as shown in the test loss spikes for Line-only, Circle-only, and Square-only runs. Additionally, world models operating over training states with a greater number of lines display higher test performance. This behavior may indicate that models trained over more diverse training data learn representations which are more resistant to perturbations.

Results showcase how in Powderworld, as more diverse data is created from the same set of core rules, world models increase in generalization capability.

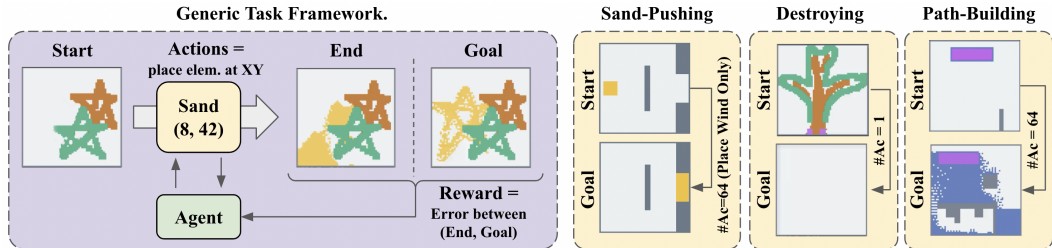

Figure 7: **In Powderworld RL tasks, agents must iteratively place elements (including directional wind) to transform a starting state into a goal state.** Within this framework, we present three RL tasks as shown above. Each task contains many challenges, as starting states are randomly generated for each episode. Agents are evaluated on test states that are unseen during training.

### 4.1.3 DOES ENVIRONMENT RICHNESS INFLUENCE TRANSFER TO NOVEL INTERACTIONS?

While a perfect world model will always make correct predictions, there are no guarantees such models can learn new dynamics. This experiment tests the *adaptability* of world models, by examining if they can quickly fine-tune on new elemental reactions.

Powderworld's ruleset is also of importance, as models will only transfer to new elements if all elements share fundamental similarities. Powderworld elements naturally share a set of behaviors, e.g. gravity, reactions-on-contact, and velocity. Thus, this experiment measures whether Powderworld presents a rich enough simulation that models can generalize to new *rules* within the environment.

To run the experiment, distinct world models are trained on distributions containing a limited set of elements. The 1-element model sees only sand, the 2-element sees only sand and water, the 3-element sees sand, water, and wall, and so on. Worlds are generated via the same procedural generation algorithm, specifically up to 5 lines are drawn. After training for the standard 5000 iterations, each world model is then fine-tuned for 100 iterations on a training distribution containing three held-out elements: gas, stone, and acid. The world model loss is then measured on a new environment containing only these three elements.

Figure 6 (top-right) highlights how world models trained on increasing numbers of elements show greater performance when fine-tuned on a set of unseen elements. These results indicate that world models trained on richer simulations also develop more robust representations, as these representations can more easily be trained on additional information. Powderworld world models learn not only the core rules of the world, but also general features describing those rules, that can then be used to learn new rules.

## 5   REINFORCEMENT LEARNING TASKS

Reinforcement learning tasks can be defined within Powderworld via a simple framework, as shown in Figure 7. Agents are allowed to iteratively place elements, and must transform a starting state into a goal state. The observation space contains the Powderworld world state $W \in \mathbb{R}^{64 \times 64 \times 20}$, and the action space is a multi-discrete combination of $X, Y, Element, V_x, V_y$. $V_x$ and $V_y$ are only utilized if the agent is placing wind.

Tasks are defined by a function that generates a starting state, a goal state, and any restrictions on element placement. Note that Powderworld tasks are specifically designed to be stochastically diverse and contain randomly generated starting states. Within this framework, many task varieties can be defined. This work considers:

- **Sand-Pushing.** The Sand-Pushing environment is an RL environment where an agent must move sand particles into a goal slot. The agent is restricted to only placing wind, at a controllable velocity and position. By producing wind, agents interact with the velocity field, allowing them to push and move elements around. Wind affects the velocity field in a 10x10 area around the specified position. Reward equals the number of sand elements

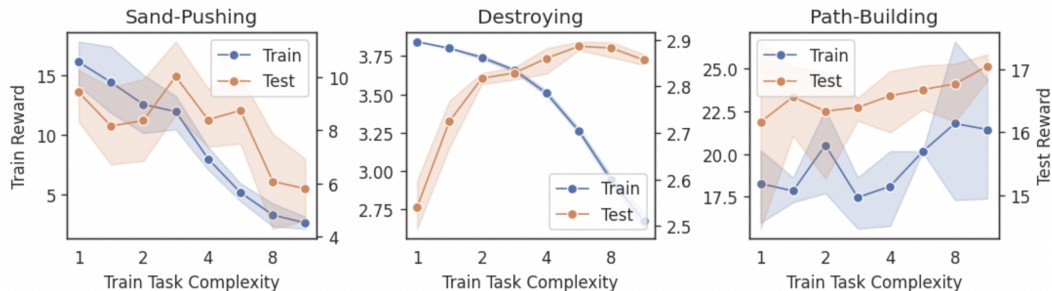

Figure 8: **Increasing the complexity of RL training tasks helps generalization, up to a task-specific inflection point.** Shown are the test rewards of RL agents trained on tasks with increasing numbers of shapes (shown in log-scale). In Sand-Pushing, too much complexity will decrease test performance, as agents become unable to extract a sufficient reward signal. In Destroying, complexity consistently increases test performance. While increased complexity generally increases the difficulty of training tasks and reduces reward, in Path-Building certain obstacles can be used to complete the goal, improving training reward.

within the goal slot, and episodes are run for 64 timesteps. The Sand-Pushing task presents a sparse-reward sequential decision-making problem.

- **Destroying.** In the Destroying task, agents are tasked with placing a limited number of elements to efficiently destroy the starting state. Agents are allowed to place elements for five timesteps, after which the world is simulated forwards another 64 timesteps, and reward is calculated as the number of empty elements. A general strategy is to place fire on flammable structures, and place acid on other elements to dissolve them away. The Destroying task presents a task where correctly parsing the given observation is crucial.

- **Path-Building.** The Path-Building task presents a construction challenge in which agents must place or remove wall elements to route water into a goal container. An episode lasts 64 timesteps, and reward is calculated as the number of water elements in the goal. Water is continuously produced from a source formation of Cloner+Water elements. In the Path-Building challenge, agents must correctly place blocks such that water flows efficiently in the correct direction. Additionally, any obstacles present must be cleared or built around.

To learn to control in this environment, a Stable Baselines 3 PPO agent (Raffin et al., 2021; Schulman et al., 2017) is trained over 1,000,000 environment interactions. The agent model is comprised of two convolutional layers with feature size 32 and 64 and kernel size of three, followed by two fully-connected layers. A learning rate of 0.0003 is used, along with a batchsize of 256. An off-the-shelf RL algorithm is intentionally chosen, so experiments can focus on the impact of training tasks.

Figure 9 highlights agents solving the various RL tasks. Training tasks are generated using the same procedural generation algorithm as the world-modelling experiments. Task-specific structures are also placed, such as the goal slots in Sand-Pushing and Path-Building, and initial sand/water elements.

To test generalization, agents are evaluated on test tasks that are out of distribution from training. Specifically, test tasks are generated using a procedural generation algorithm that only places squares (5 for Destroying and Sand-Pushing, 10 for Path-Building). In contrast, the training tasks are generated using only lines and circles.

Figure 8 showcases how training task complexity affects generalization to test tasks. Displayed rewards are averaged from five independent training runs each. Agents are trained on tasks generated with increasing numbers of lines and circles (0, 1, 2, 4 ... 32, 64). These structures serve as obstacles, and training reward generally decreases as complexity increases. One exception is in Path-Building, as certain element structures can be useful in routing water to the goal.

Different RL tasks display a different response to training task complexity. In Sand-Pushing, it is helpful to increase complexity up to 8 shapes, but further complexity harms performance. This inflection point may correspond to the point where learning signal becomes too high-variance. RL

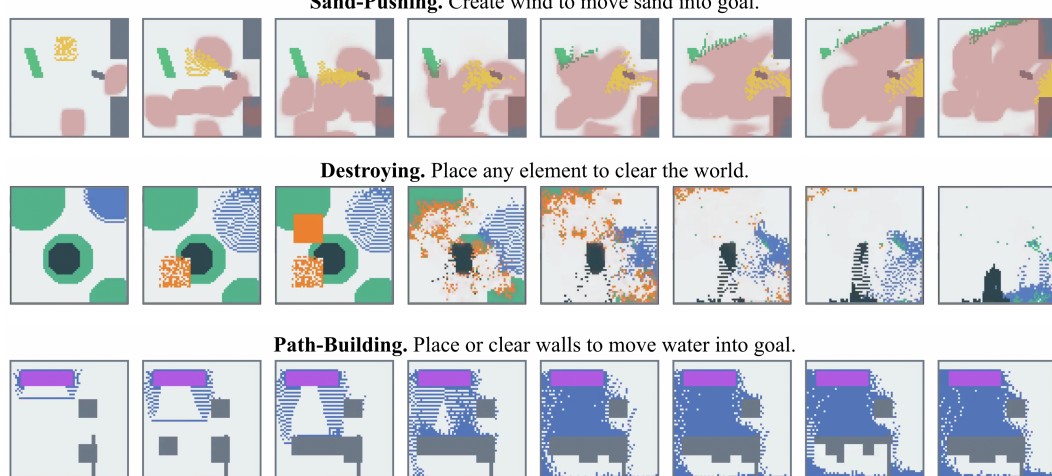

Figure 9: **Agents solving the Sand-Pushing, Destroying, and Path-Building tasks.** In the Sand-Pushing task, wind is used to push a block of sand elements between obstacles to reach the goal slot on the right. In Destroying, agents must place a limited number of elements to efficiently destroy the world. In Path-Building, agents must construct a path for water to flow from a source to a goal container. Tasks are randomly generated via a procedural algorithm.

is highly dependent on early reward signal to explore and continue to improve, and training tasks that are too complex can cause agent performance to suffer.

In contrast, agents on the Destroying and Path-Building task reliably gain a benefit from increased training task complexity. On the Destroying task, increased diversity during training may help agents recognize where to place fire/acid in test states. For Path-Building, training tasks with more shapes may present more possible strategies for reaching the goal.

The difference in how complexity affects training in Powderworld world-modelling and reinforcement learning tasks highlights a motivating platform for further investigation. While baseline RL methods may fail to scale with additional complexity and instead suffer due to variance, alternative learning techniques may better handle the learning problem and show higher generalization.

## 6  CONCLUSION

Generalizing to novel unseen tasks is one of the grand challenges of reinforcement learning. Consistent lessons in deep learning show that *training data* is of crucial importance, which in the case of RL is training tasks. To study how and when agents generalize, the research community will benefit from more expressive foundation environments supporting many tasks arising from the same core rules.

This work introduced Powderworld, an expressive simulation environment that can generate both supervised and reinforcement learning task distributions. Powderworld's ruleset encourages modular interactions and emergent phenomena, resulting in world models which can accurately predict unseen states and even adapt to novel elemental behaviors. Experimental results show that increased task complexity helps in the supervised world-modelling setting and in certain RL scenarios. At times, complexity hampers the performance of a standard RL agent.

Powderworld is built to encourage future research endeavors, providing a rich yet computationally efficient backbone for defining tasks and challenges. The provided experiments hope to showcase how Powderworld can be used as a platform for examining task complexity and agent generalization. Future work may use Powderworld as an environment for studying open-ended agent learning, unsupervised environment design techniques, or other directions. As such, all code for Powderworld is released online in support of extensions.

ACKNOWLEDGMENTS

This work was supported by a Packard Fellowship to P.I. Thanks to Akarsh Kumar for assistance during discussions, paper feedback, and implementation on the RL tasks.

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

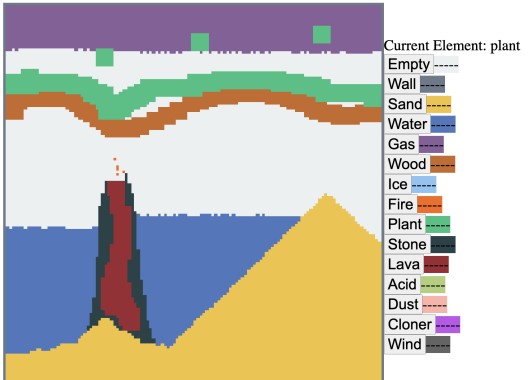

Figure 10: **The Powderworld Web GUI allows users to directly edit the state of a world by drawing on various elements.** World states can be generated at custom resolutions, and the simulation runs in real-time. Powderworld runs on a GPU server, with the web app acting only as a display and controller.

## A  APPENDIX

### A.1  POWDERWORLD ENGINE

Described in this section is an overview of the technical details of the Powderworld Engine. We provide these details 1) as a reference for future work built on top of Powderworld, and 2) as a framework for building simulations and environments that run on the GPU. Powderworld is designed to run as fast as possible on a standard GPU environment, as many deep learning setups already support GPU access. Thus, Powderworld interacts with the GPU through Pytorch functions.

**Data Representation.** The state in Powderworld is represented as a BxHxWxN size tensor. One benefit of running on the GPU is that multiple simulations can be run in parallel, therefore all functions in Powderworld are written to support *batches* of data. The other dimensions refer to height and width, along with an N-size vector representing the data stored in each location. This data is specifically a one-hot vector of the current element at the location, along with indices dictating gravity, density, flow state, and velocity.

**Matrix Operations.** To remain efficient and parallelizable, operations in Powderworld are not implemented as loops over XY space, but rather as series of matrix operations. For example, to simulate the world falling down one block, one can call the following function:

```
world[:] = torch.roll(world, shifts=1, dims=2)
```

In the codebase, there are helper functions for shifting the world each of four directions: getBelow, getAbove, getLeft, getRight. These functions are called frequently and used to build up more complex operations.

Additionally, since operations need to occur over the entire world matrix, to modify specific portions of the matrix we must construct a mask. We can do this by doing any kind of comparison over values in the world matrix, then setting the world matrix to a weighted sum of world = (world*(1-mask) + newWorld*mask). For example, to change all fire elements into water elements, the following pseudocode works:

```
// Transform fire into water.
mask = (world[:, fire_index] == 1)
world[:] = world*(1-mask) + water_vec*(mask)
```

**Gravity.** In Powderworld, each element has a Density and an IsGravity flag. These values are stored in the 1st and 2nd indicies of the world array. Gravity operates as a series of switches: if an element is above another element and the top element has greater density, and both elements are gravity-enabled, then the two elements swap. As a baseline, empty space (Air) has a density of 1,

thus elements like Sand (density=2) will fall, and elements such as Gas (density=0) will rise. The IsGravity flag is necessary to prevent elements from falling through stationary elements such as wall or wood, which should remain static. Gravity handles only vertical swaps, and in combination with other behaviors creates the piling and flowing mechanics seen in sand and water.

One crucial component of the gravity procedure is that due to the nature of switching, two elements cannot attempt to switch into the same position. Swaps are performed simultaneously, and a swap involves setting the upper position to the lower element, and vice versa. If two elements swap into the same position, then that position will be written to twice, and the original element will duplicate itself above and below. Therefore, all swap operations must be sure to never swap two elements into the same position. To solve this in the gravity case, we iterate gravity as a loop over the possible densities. If all densities were computed together, a vertical stack of densities [0,1,2] would result in the elements with densities of 0 and 2 both attempting to move into the center position. By processing downward swaps for each density iteratively, an order is established and the conflict does not occur (in the example provided, 0 would first swap with 1, and then 0 would swap with 2 in the following iteration.

```
// Run gravity.
for currDensity in [0,1,2,3]:
{
density = world[:, density_index]
# Delta between ABOVE and current
density_delta = get_above(density) - density
is_density_above_greater = (density_delta > 0)
# If BELOW has density_above_greater, then density_below_less
is_density_below_less = get_below(is_density_above_greater)
is_density_current = (density == currDensity)
is_density_above_current = get_above(is_density_current)
is_gravity = (world[:, gravity_index] == 1)
is_center_and_below_gravity = get_below(is_gravity) & is_gravity
is_center_and_above_gravity = get_above(is_gravity) & is_gravity

world_above = get_above(world)
world_below = get_below(world)
world[:] = world[:]*(1-does_become_below-does_become_above)
    + world_below*does_become_below + world_above*does_become_above
}
```

**Sand-Piling.** Both the sand and dust elements display additional behavior when falling. These elements cannot support themselves upright, and if there is an empty space to their bottom-left or bottom-right then the element will fall into that location. In practice, this means that the sand elements form stable pyramids and hills.

To implement the sand-piling behavior, we check if each element is a sand-type (sand or dust), then if either the bottom-left or bottom-right space is a lower density, the two elements swap. To prevent ambiguity from left/right falling and break symmetry, a random value is sampled for each position dictating whether to check bottom-left first or bottom-right.

**Water-Flowing.** Water and other fluids (water, lava, acid) behave similarly to sand, except that these elements can also flow left and right. This behavior is implemented in a similar fashion to sand-piling, except that the left and right adjustment elements are considered for swapping, instead of bottom-left and bottom-right.

In addition, an optimization is implemented to increase the effective velocity of fluids. With the naive setup, each fluid can flow either left or right. However, this means that large clumps of fluids often have a hard time spreading out, as they rely on random osmisis in order to fully spread horizontally. To speed up the process, an index is reserved to keep track of the direction that a given fluid element has previously flowed in. If a fluid has previously flowed to the left, then the next timestep it will first check if it can move to the left (rather than randomly selecting left/right). This ruleset means that a single particle of water will continue flowing left until it hits a wall, rather than randomly move between left/right, creating a more dynamic fluid system.

**Ice and Water.** The goal with ice and water are to create two elements with different phases (solid and liquid), which transition between one another depending on their number of neighbors. Ice has a default chance of melting into water (2% a timestep), and water has a chance of freezing into ice if it has three or more ice neighbors (5% a timestep).

To compute the number of neighbors for this behavior and more, a simple convolutional kernel is employed. The kernel has a 3x3 kernel size and contains all ones, resulting in each position after running the convolution containing the number of a specific element that exist next to that position.

```
self.neighbor_kernel = torch.ones((1, 1, 3, 3), device=device)
water_can_freeze = (F.conv2d(self.get_elem(world, "ice"),
    self.neighbor_kernel, padding=1) >= 3)
does_turn_ice = self.get_bool(world, "water") & water_can_freeze
    & (ice_chance < 0.05)
world[:] = interp(switch=does_turn_ice, if_false=world,
    if_true=self.elem_vecs['ice'])
```

**Fire-Burning.** Fire is implement as an element which cannot survive on its own. Fire always has a chance to burn away into air. However, fire that is next to a burnable element (wood, plant, dust) will not burn away, and instead has a chance to transform that element into fire. Thus, fire will travel along the paths of burnable elements, setting fire to anything close enough to touch. Fire also naturally moves upwards, thus fire can jump from one element to another even across a small air gap.

Note that each burnable element has a distinct behavior when burned. Wood burns the slowest, as it has the lowest chance of burning when in contact with a fire element. Plant burns faster, and dust burns the fastest and additionally creates large amounts of velocity when burned. This aims to simulate an explosive effect as the velocity will scatter nearby particles outwards.

**Plant-growing.** Plants spread in water, but in an incomplete fashion. The intent is to create a vine-like structure where plants absorb water and create more plant, leaving behind a web of plant elements. Specifically, water elements that are near greater than four plants have a chance to turn into either water or air.

**Lava.** Lava is a liquid that flows similarly to water. Lava that is exposed to air continuously creates fire at those positions, thus lava acts as a constant source of fire that does not naturally dissipate. As lava obeys gravity and fluid flowing, lava can flow down structures and reach new locations. Lava that comes into contact with water forms a stone element in that location.

**Acid.** Acid is a fluid which destroys other elements. Specifically, if acid is neighboring a non-empty element, then there is a 20% that the acid block will dissapear along with any neighboring elements. Normally, elements should not be able to interact with all its neighbors simultaneously (may cause conflicts), but since acid only destroys things, there is no issue.

**Cloner.** The final element is a cloner, which keeps track of the first element that touches it, then continuously produces that element at any adjacent empty positions. Cloner elements are meant to be used as a source of mobile elements such as water or gas, and once assigned on contact can create structures such as waterfalls or spouts.

**Velocity System.** Powderworld elements interact with each other via elemental reactions as well as a global velocity system. Each position in the world holds an x and y velocity, and blocks move if the magnitude of the velocity at that position is greater than a threshold. The moving behavior consists of a loop over the eight primary directions, and first checks if the velocity at each location is aligned in that direction. Then, assuming the velocity magnitude is great enough, the procedure checks if there is an empty space in that direction, and if so, the element moves. All elements are effected by velocity except for walls.

Additionally, the velocity layer also goes through its own simulation. While other powder games use fluid dynamics to simulate velocity, this work instead opts to use a simpler but quicker method. Specifically, velocities create additional velocity in the direction they point in. This is done during the same loop as above. Next, velocities are slightly averaged with their neighbors, and the entire velocity layer is scaled down by a factor of 0.95. Overall, the velocity simulation allows velocity values to travel forwards in the direction they point in, while slightly spreading out and decaying.

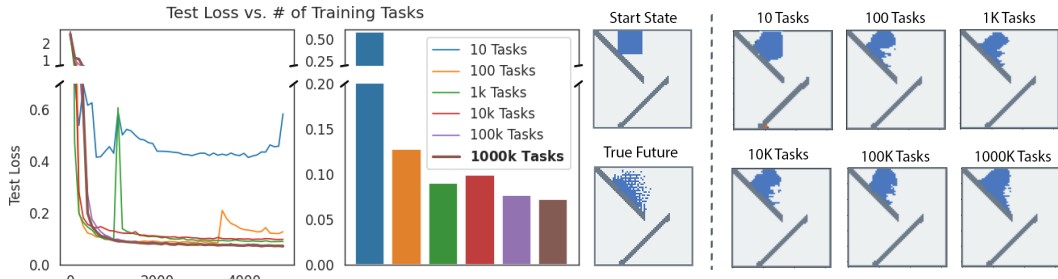

Figure 11: **World model generalization improves as the number of starting states is increased.** Shown are the test performances of world models trained with data from 10 states, 100 states, 1000 states, etc. Models trained on less data show greater instability, as observed by the spikes in test loss. Right: Comparison of sampled world model predictions on a test state.

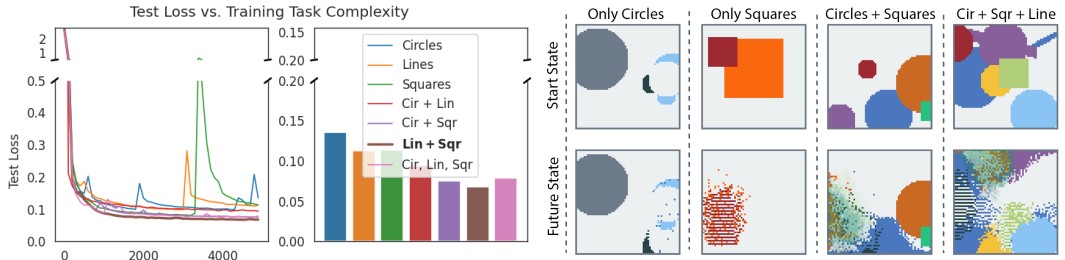

Figure 12: **Increasing environment complexity by including additional shapes improves transfer performance.** World models trained on tasks including lines, circles, and squares create diversity, enabling generalization to unseen tasks. Right: Sampled tasks generated with varying types of shapes.

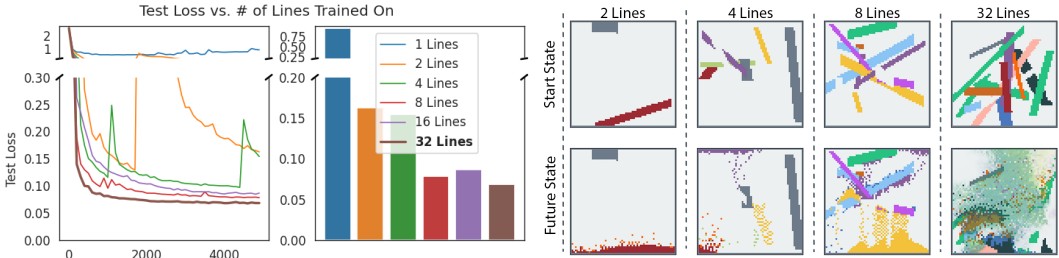

Figure 13: **Training on environments with more lines results in stronger generalization.** A higher number of lines increases the diversity of training states, but may also create destructive reactions.

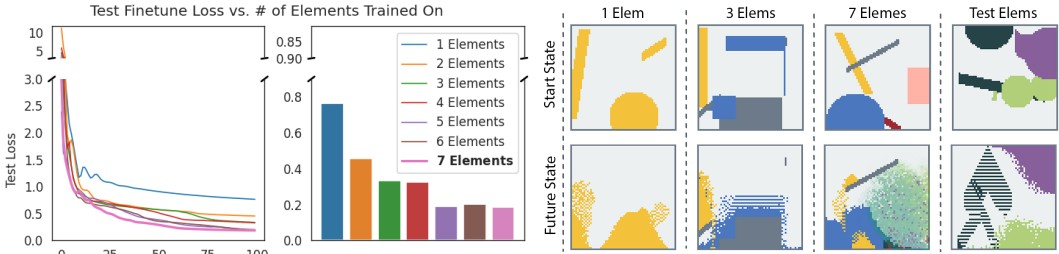

Figure 14: **World models trained on more elements showcase better performance when fine-tuned on novel elements.** When transferring to an environment containing three held-out elements (gas, stone, acid), models exposed to more elements during training perform better. These results show that Powderworld provides a rich enough simulation that world models learn robust representations capable of adaptation. The richer the simulation, the stronger these representations become.

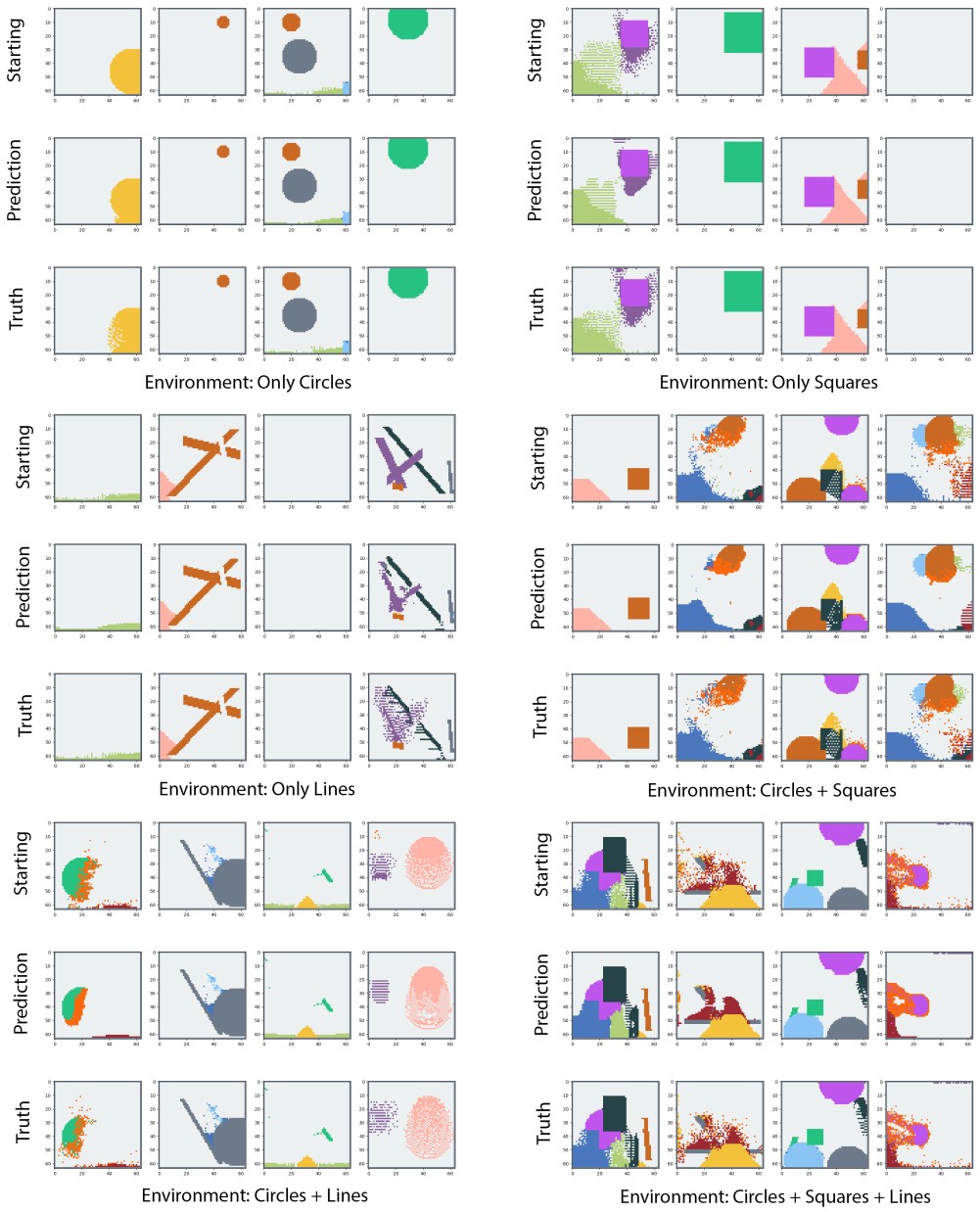

Figure 15: **Training tasks at various environment complexities.** Environment complexity is defined as the number of shapes included within the procedural generation algorithm. Each shape is generated up to five times, at random positions, sizes, and with a random element.

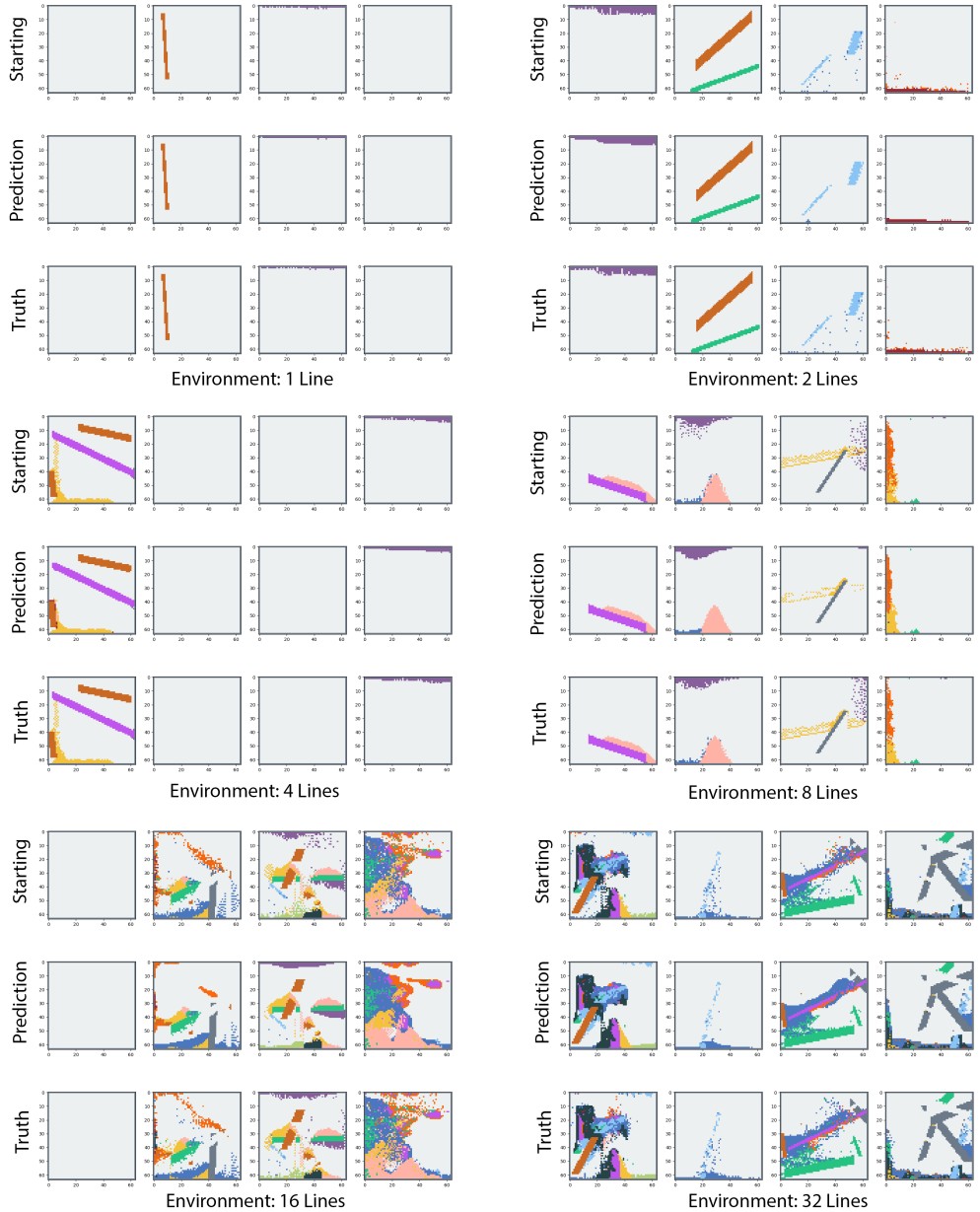

Figure 16: **Training tasks at various numbers of lines.** Note that each line number represents the maximum possible number of lines, thus the 4-Line environment can generate [0,1,2,3,4] lines. Blank worlds appear when no lines are generated, or lines are generated out of unstable elements (e.g. fire) that disappear over time.

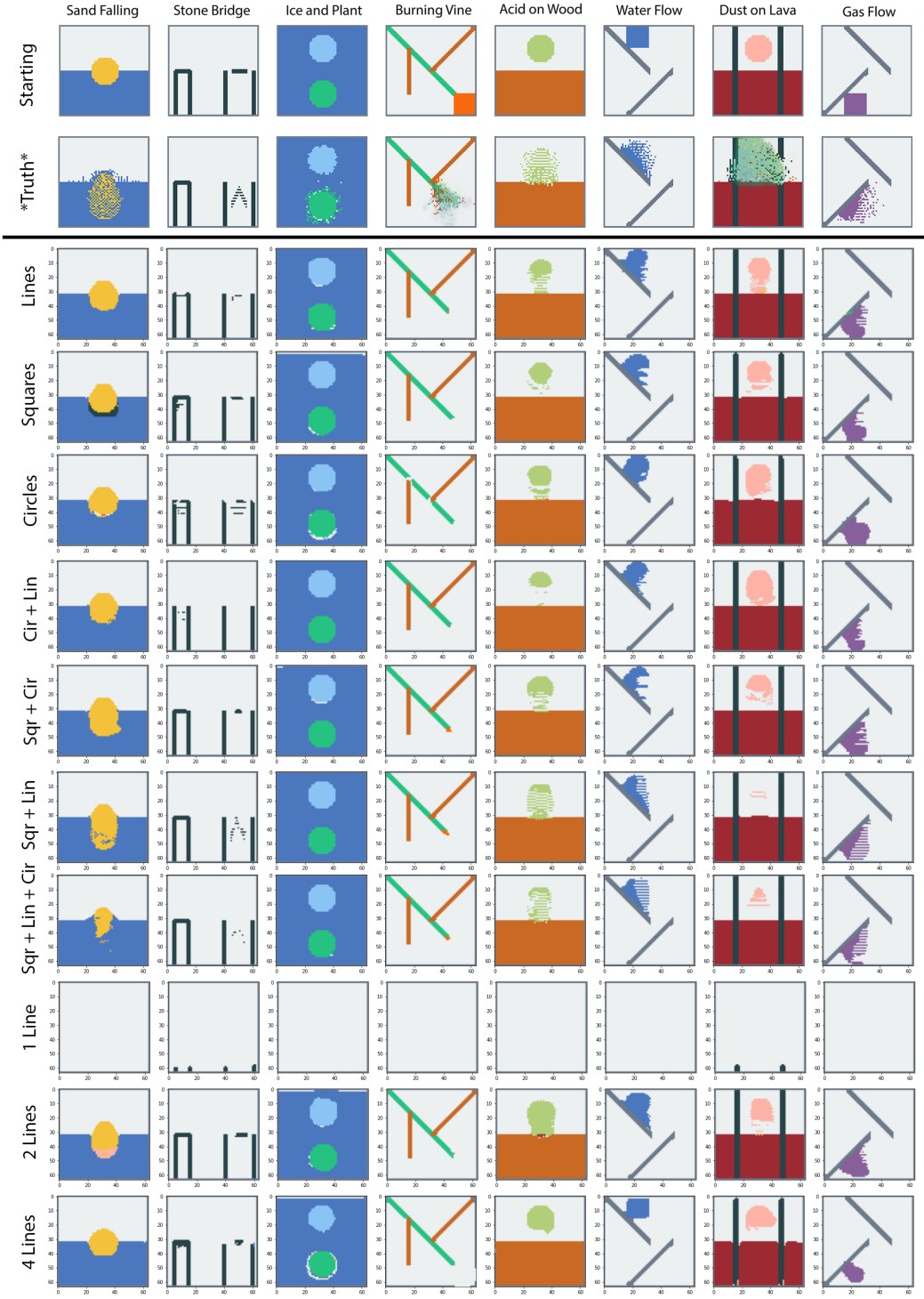

Figure 17: **World model predictions over test tasks.** This figure showcases the eight test tasks simulated for 16 timesteps into the future. World models are trained to predict 8 timesteps forwards, thus results are shown with each world model applying two updates.

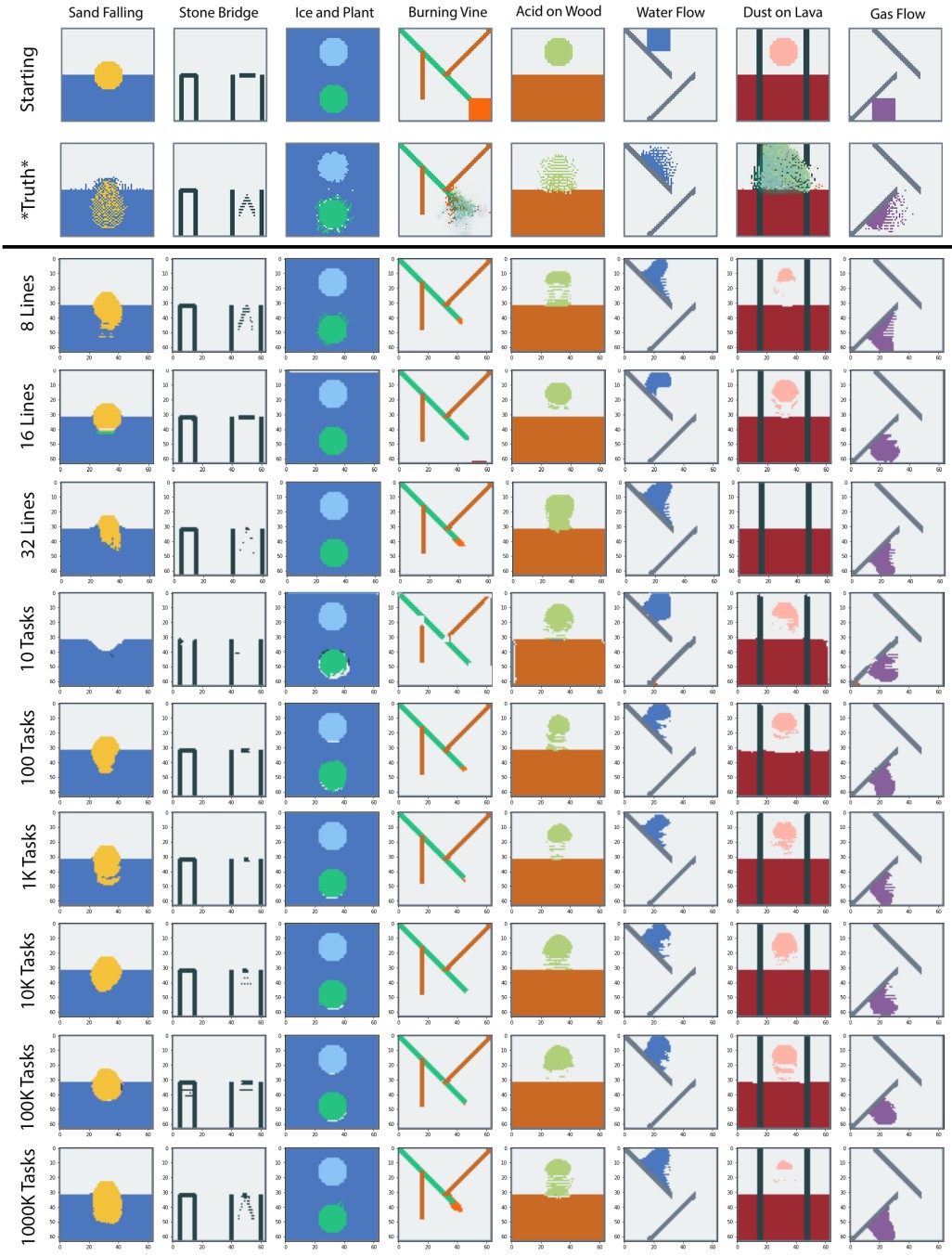

Figure 18: **Cont: World model predictions over test tasks.**

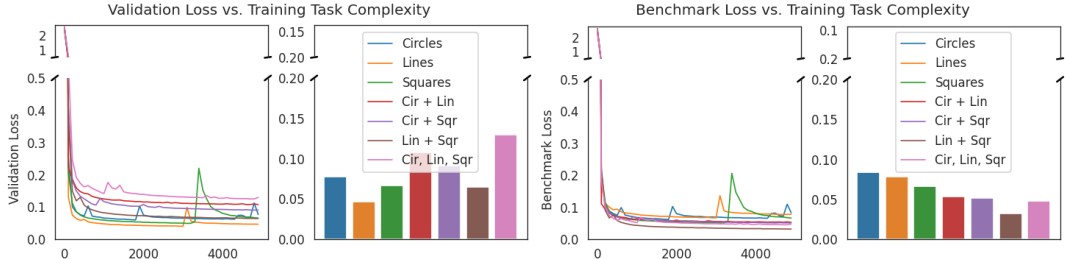

Figure 19: **World model training curves on environments with increasing complexity.** The Validation loss represents the performance of the world models on tasks sampled from their training distribution (i.e. only circle, only squares, etc.) Note that these tasks are never actually seen during training. The loss represents performance over an arbitrarily-chosen set of tasks, specifically, worlds with 5 lines.

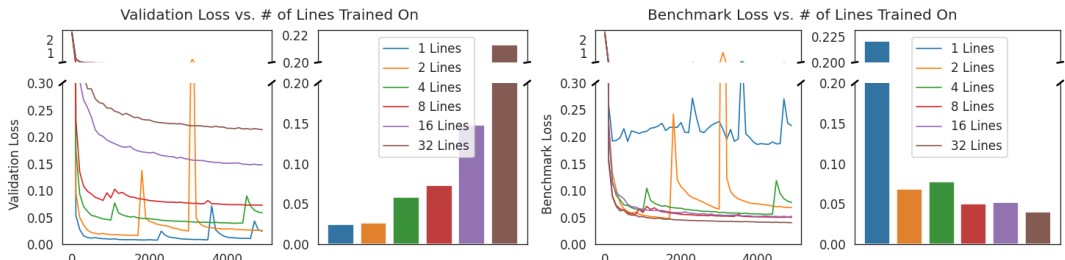

Figure 20: **World model training curves on environments with increasing number of lines.** Note the mirrored correlations: world models trained on more complex environments show higher validation loss (as the tasks are harder) but lower benchmark loss.

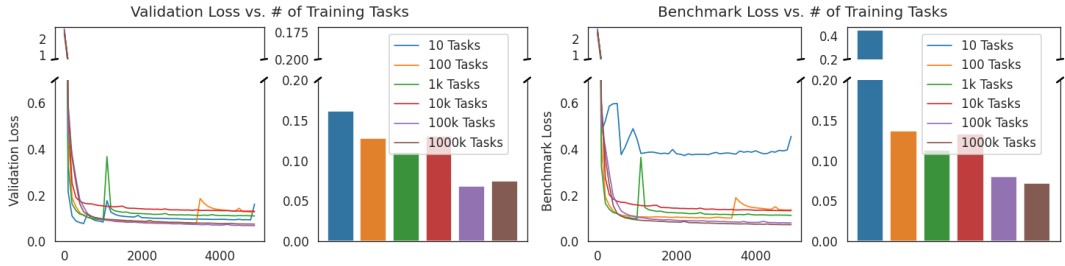

Figure 21: **World model training curves on increasing number of training tasks.** More training tasks consistently improves both Validation and Benchmark performance.

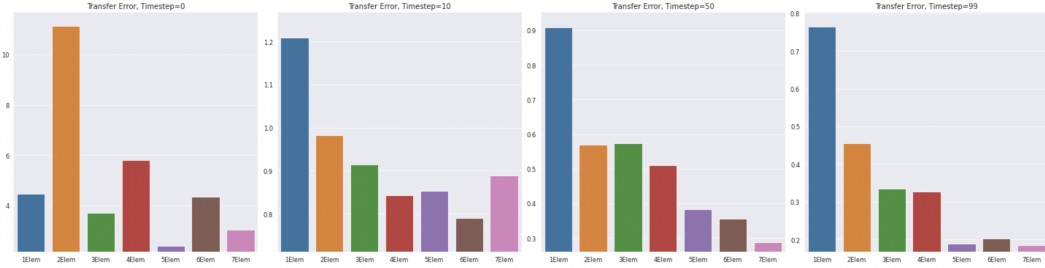

Figure 22: **Transfer performance to novel elements, over fine-tune time.** Showcased are world models trained on increasing numbers of elements, then fine-tuned on an environment with three novel elements (gas, stone, acid). While in the zero-shot setting there is little correlation in performance, fine-tuning reveals that models trained on larger numbers of elements can more efficiently adapt to the new environment.

