# OpenReview forum: "Powderworld: A Platform for Understanding Generalization via Rich Task Distributions"
_ICLR.cc/2023/Conference — ICLR 2023 notable top 25%_

### Official Review · Reviewer_oDrU · 2022-10-17

**Confidence:** 4
**Correctness:** 3
**Technical Novelty And Significance:** 2
**Empirical Novelty And Significance:** 2
**Recommendation:** 8

**Clarity, Quality, Novelty And Reproducibility:**

The paper is clear and supposedly reproducible with open-source code forthcoming. The novelty is more challenging. How is this more useful than Crafter/MiniHack?

**Strength And Weaknesses:**

### Strengths
- The paper is well-written and well-motivated, as training agents on large, diverse task distributions has become increasingly popular and important as we seek more general agents.
- The environment seems very fast, given it runs on a GPU, which is great for iteration speed for research.

### Weaknesses
- The world modelling task is definitely interesting but it is hard to see how it is directly relevant outside of this environment. We would likely never have access to a Markovian state in such a controlled setting. The section appears to be motivated by works such as World Models and Dreamer, but in those cases 1) the models are learned directly from pixels without a Markovian state 2) there is an agent taking actions in the world. So this is a totally different paradigm. The fact that the model generalizes better with more data here is expected, as the authors note this has been the case in a variety of other settings already.
- How can we be sure the hand designed tasks are unbiased? For all we know they could be somewhat arbitrary.
- While the motivation in the intro is that this world is more general than others such as MiniGrid/Crafter/MiniHack, the only RL task presented is just sand pushing. How is this more diverse and useful than for example the tasks in Crafter/MiniHack which vary from navigation to tool use?
- It looks like the experiments were all just one seed. When we know RL training is volatile, it seems like an oversight to have done this given the environment is meant to be fast.
- One of the motivations in the intro is the potential use for UED, but there is no demonstration of this. It would be interesting to see if this environment offers something unique here vs. the alternatives. It may be beyond the scope to run this for a rebuttal but it would likely see an increased score.

**Summary Of The Paper:**

This paper presents Powderworld, a fast simulated environment capable of producing diverse tasks for either supervised learning or an RL agent. The paper demonstrates that Powderworld can be used for both learning world models and training RL agents in a sandpushing task.

**Summary Of The Review:**

At the surface the paper presents an interesting and unique environment that may facilitate future research. However, the two presented use cases are a world modelling task which is not directly relevant outside powderworld, and an RL task where the main result confirms what we already know from Procgen et al. Overall I do not think the paper contributes anything new at present.

Post discussion:

The paper is completely revised so now provides some useful, fast settings that can be used for work in PCG and UED for RL. Analysis shows this can be a worthy testbed and may help researchers enter what is a growing and important field.

---

> ### Author Response · Authors · 2022-11-18
> **Response to oDrU**
>
> Thank you for your useful feedback. We have taken this feedback to heart and thoroughly updated the RL experimental section of the paper. While the Powderworld environment remains as described, we have trained a range of RL agents on three novel tasks (sand-pushing, destroying, path-building) and demonstrate each agent's generalization capability to unseen test tasks.
>
> The updated sections can be seen in Figures 7,8,9. We also present videos of the RL tasks at: https://resourceful-human.static.app/  We believe that the updated paper presents sufficient results to motivate the environment -- if you agree, would you consider increasing your score? We are happy to present additional experiments if they are helpful.

---

> > ### Author Response · Authors · 2022-11-18
> > **Response (Part 2)**
> >
> > Some specific comments below:
> >
> > >The world modelling task is definitely interesting but it is hard to see how it is directly relevant outside of this environment. We would likely never have access to a Markovian state in such a controlled setting. The section appears to be motivated by works such as World Models and Dreamer, but in those cases 1) the models are learned directly from pixels without a Markovian state 2) there is an agent taking actions in the world. So this is a totally different paradigm. The fact that the model generalizes better with more data here is expected, as the authors note this has been the case in a variety of other settings already.
> >
> > You are correct, and we argue that the world model sections presents a sanity-checking benchmark. Specifically, world model experiments are presented to highlight:
> > -   **Generalization improves with more training data**. This is an expected result, presented to confirm that Powderworld-simulated data displays similar scaling properties as real-world data.
> > -   **Generalization improves with more complex training states**. These experiments show that the _diversity_ of training states matters. These results motivate the benefit of multi-task learning, and encourage research on methods such as UED that automate task design.
> > -   **Models generalize to NEW elements.** We argue that this result is substantially novel and unique to Powderworld. Because of the modular nature of Powderworld, we can examine models trained on a subset of the ruleset (1 element, 2 elements, etc.). Because models trained on more elements show better transfer performance to unseen elements, it shows that models succesfully learn not only specific rules (i.e. fire burns wood), but also general properties of Powderworld _as a system_ (i.e. elements are affected by gravity, elements only affect local neighbors, etc).
> >
> >
> > > How can we be sure the hand designed tasks are unbiased? For all we know they could be somewhat arbitrary.
> >
> > This is a fair point. To decrease bias due to hand-design, when evaluating reinforcement learning agents, **we update the paper to instead generate test tasks using a unique PCG algorithm.**  Specifically, training tasks are generated via lines/circles, while test tasks are generated using squares.
> >
> > >While the motivation in the intro is that this world is more general than others such as MiniGrid/Crafter/MiniHack, the only RL task presented is just sand pushing. How is this more diverse and useful than for example the tasks in Crafter/MiniHack which vary from navigation to tool use?
> >
> > We have updated the experimental section to present three new RL tasks within Powderworld, along with a general framework for definiing RL task varieities (Figures 7,8,9). Videos of these tasks can be seen at the anonymized site: https://resourceful-human.static.app/ .
> >
> > > It looks like the experiments were all just one seed. When we know RL training is volatile, it seems like an oversight to have done this given the environment is meant to be fast.
> >
> > The RL experiments were run with 5 seeds; although this was not properly communicated in the original paper. We have updated the new RL results section to present error bars representing variance between the five seeds.
> >
> > In addition, we have updated the paper to consider three RL tasks rather than only sand-pushing, to demonstrate the diversity of Powderworld tasks.
> >
> > >One of the motivations in the intro is the potential use for UED, but there is no demonstration of this. It would be interesting to see if this environment offers something unique here vs. the alternatives. It may be beyond the scope to run this for a rebuttal but it would likely see an increased score.
> >
> > You are correct; and we agree that Powderworld was designed to support UED methods. To properly present and study UED methods on Powderworld would require large experiments, and we consider this in the scope of future work. However, here are the design reasons we believe Powderworld is suited for UED study:
> > - Tasks can be defined by simple matricies (a start state and goal state). We've updated the RL section to properly showcase how new tasks can be created within Powderworld.
> > - We've shown that training task complexity helps increase generalization. These results show that the PCG parameters used to create training tasks is important -- the next extension beyond "modify PCG parameters" is to "make new PCG algorithms", which is the realm of UED.
> >
> > >How is this more useful than Crafter/MiniHack?
> >
> > Crafter and MiniHack are both great environments. Powderworld is presented as middle ground between these lightweight tasks and more complicated simulations such as Minecraft. We argue that Powderworld is useful as 1) it is fast to run, 2) complex dynamics can be seen within a small # of simulation timesteps, and 3) it is simple to define additional tasks based on shared Powderworld rules.

---

> > > ### Comment · Reviewer_oDrU · 2022-11-19
> > > **Paper now significantly better :)**
> > >
> > > Hi authors,
> > >
> > > The title is a spoiler, but the paper is now clearly a lot stronger. The RL experiments have gone from being a rushed single seed in a single setting, to multiple domains with multiple seeds and new analysis provided of agent behaviors and generalization capabilities. I still believe the world modelling task is not as useful, but maybe it can be a sanity check for video prediction work elsewhere.
> > >
> > > I still completely stand by my original review, but the paper is now drastically better and so if I was reviewing it now from scratch it would be an accept. So, here is a 3x score increase.
> > >
> > > Good work!

---

### Official Review · Reviewer_fLgh · 2022-10-25

**Confidence:** 4
**Correctness:** 4
**Technical Novelty And Significance:** 4
**Empirical Novelty And Significance:** 4
**Recommendation:** 8

**Clarity, Quality, Novelty And Reproducibility:**

The paper is very clear. The authors provide a lot of comments and intuitions about their experimental results. The work is novel.  Code is provided so experiments should be reproducible.


**Strength And Weaknesses:**

### Strengths

The platform supports the design of environments with different levels of complexity and many degrees of freedom (i.e. many different elements can be added to the environment).

It can also run many instantiations (multiple worlds) in parallel on a GPU, which translates into runtime efficiency.

Environments can be either manually or procedurally generated, which means that one can quickly generate thousands of different worlds to train on.

The environments are also modular in the sense that the different elements interact only locally with their neighbors.

The paper provides extensive experimentation, especially for the world model case.  The results are intuitive and generally agree with our intuitions of how adding more, tasks and/or complexity increases generalization.

### Weaknesses

According to the paper, the interactions between elements are computed via convolutional matrix multiplications. This means that the transition dynamics can be model **exactly** with a convolutional neural network (CNN). That is, theoretically, a CNN can be trained to zero loss, provided you can find the global optimum. An increase in complexity (i.e. adding more elements to the environment) can be resolved by adding more kernels to the CNN such that each kernel can potentially model each of the possible interactions. Can the authors comment on this? Or even better, provide results showing whether zero loss can be obtained for simple environments. Note that this won’t affect my score but I am curious to know if this is the case.

Since the main purpose of this paper is training RL agents I am missing more experiments showing what can be done with this platform. So far I can only think of tasks like moving objects (elements) from one place to another or removing ceratin elements. Can the authors provide examples of other tasks?

Related to my previous question, have you considered adding agents (e.g. robots) that can interact with the elements?

In the last RL experiment, you mention that increasing the complexity of the environments decreases generalization performance. Have you tried to start with simple environments and gradually increase the complexity as the agent improves its policy?

In the first reference, there is a link to a site where there is a game called powder game. Is this the Powderworld platform?

Perhaps this question is too technical but, how do you take care of the non-linear operations (e.g. gravity), how can those be performed on a GPU?


**Summary Of The Paper:**

This paper presents Powderworld, a platform to build environments on which to train world models and RL agents. The main purpose of this platform is to investigate generalization.


**Summary Of The Review:**

This is a strong paper introducing a new platform for designing RL environments. I believe Powederworld can be very valuable to the community.

---

> ### Author Response · Authors · 2022-11-18
> **Response to Reviewer fLgh**
>
>  > According to the paper, the interactions between elements are computed via convolutional matrix multiplications. This means that the transition dynamics can be model exactly with a convolutional neural network (CNN). That is, theoretically, a CNN can be trained to zero loss, provided you can find the global optimum. An increase in complexity (i.e. adding more elements to the environment) can be resolved by adding more kernels to the CNN such that each kernel can potentially model each of the possible interactions. Can the authors comment on this? Or even better, provide results showing whether zero loss can be obtained for simple environments. Note that this won’t affect my score but I am curious to know if this is the case.
>
> Thank you for raising this point. The element interactions are indeed computed via a convolutional kernel, however, these kernels contain non-linear functions. For example, the behavior of an ice block is to turn into water if less than three of its neighbors are ice. This can be seen as a 3x3 convolutional kernel summing up the number of ice numbers, followed by a step function if the total is < 3. Thus, methods using a differentiable model to approximate these dynamics may not be able to exactly replicate dynamics.
>
> > Since the main purpose of this paper is training RL agents I am missing more experiments showing what can be done with this platform. So far I can only think of tasks like moving objects (elements) from one place to another or removing ceratin elements. Can the authors provide examples of other tasks?
>
> **We've thoroughly updated the paper to provide more tasks, along with a general Powderworld task framework.** Specifically, tasks within Powderworld can be specified via a start state and a goal state. Agents can take actions that place elements at XY positions. See Figures (7,8,9).
>
> In this revised framework, a variety of tasks can be defined. We specifically present three additional task distributions:
> - "Sand-Pushing", in which an agent must place wind that moves sand into a goal.
> - "Destroy this", in which the agent attempts to clear as much of the state as possible (by placing lava, fire, acid, etc.)
> - "Path-Building", in which an agent places/removed wall elements to guide flowing water into a container.
>
> We also show that the generalization experiments show differing results on these tasks. On the Destroying and Path-Building tasks, test generalization scales well with complexity, while on the Sand-Pushing tasks, too much complexity will harm performance. These tasks are presented as a starting point for further work ontop of the Powderworld benchmark.
>
> > Related to my previous question, have you considered adding agents (e.g. robots) that can interact with the elements?
>
> This is a great idea; and we've considered adding an "Embodied Powderworld" extension involving embodied agents. These agents can be defined as neural cellular automata, and can move either by hand-designed rules or by a differentiable policy. Embodied agents opens a whole box of options, and while we consider it out of the scope of this paper, it is a welcome direction for future work.
>
> > In the last RL experiment, you mention that increasing the complexity of the environments decreases generalization performance. Have you tried to start with simple environments and gradually increase the complexity as the agent improves its policy?
>
> This is another great idea, and we would love to pose this suggestion for future work. Curriculum learning is a rich field, and many methods could be applicable to increase performance on the Powderworld generalization benchmarks.
>
> > In the first reference, there is a link to a site where there is a game called powder game. Is this the Powderworld platform?
>
> The reference is to a site containing online powder games, these are not the same as Powderworld. **We have hosted Powderworld on an anonymized site:** please view https://resourceful-human.static.app/  in your browser to interact with a live Powderworld simulation.
>
> > Perhaps this question is too technical but, how do you take care of the non-linear operations (e.g. gravity), how can those be performed on a GPU?
>
> Not too technical at all! A thorough description of the operations is detailed in the appendix and provided codebase. At a high level, gravity works by performing swaps on elements with differing densities. We compute a boolean mask of elements whose densities are lower than those above them (through matrix operations on the GPU). Elements marked to swap then transform into a down/up-shifted world (equivalent to swapping two elements). We believe a major contribution of this work is a framework for building simulations through GPU/Pytorch operations, and are happy to answer any questions regarding technical details.

---

> > ### Comment · Reviewer_fLgh · 2022-12-05
> > **Response to authors**
> >
> > I want to thank the authors for their response. After reading the other reviews and author responses, I remain in favor of accepting this paper.

---

### Official Review · Reviewer_EMHS · 2022-10-28

**Confidence:** 4
**Correctness:** 3
**Technical Novelty And Significance:** 3
**Empirical Novelty And Significance:** 3
**Recommendation:** 8

**Clarity, Quality, Novelty And Reproducibility:**

Quality
  * The proposed ideas and implementation tackle an important problem. The formulation and methods are technically sound.

Clarity
  * The paper is reasonably well written but some parts are unclear. In particular, training data (trajectory) generation and test time eval for world modeling and RL might benefit from additional detail or illustration.

Novelty
  * The simulation environment seems quite novel.

Reproducibility
  * No issues here as code is released.

**Strength And Weaknesses:**

Strengths
  - Novel environment designed to facilitate study in key topic in agent design (generalization across tasks). The design makes it easy to extend the environment, vary task distributions and generate training data inexpensively.
  - Efficient design and implementation of the simulator, leveraging modern GPU hardware.

Weaknesses
  - Some parts of the experimental setup could be more clearly described. An illustration of the trajectory generation process for training and inference might help improve clarity in this part of the paper.
    - I was confused by the implemetation details and scaling law claims in Sec 4.1.1.
      - Were each of the 5 models in Fig 5 trained on the same total amount of simulation data or not? Does the phrase "training data" in the line "world models trained on increasing amounts of training data display higher performance on a set of test tasks" refer to the number of tasks while keeping the total number of simulations constant across all 5 models? If yes, how exactly were these trajectories sampled? And does it mean that Fig 5 right is a sample observation (W_15) from the predicted logit probs (W').
      - What exactly is a run and a trial? I found the statements "each run trains on 10x more data as the previous" and "Runs are trained for three trials and the average test loss over training time is displayed" confusing.
    - A similar question arises in Sec 5. How exactly are the 2.5M datapoints for the RL agent generated for agents restricted to T training tasks?
  - (More a question than a weakness) How well does powderworld support human agents? More concretely, how easy would it be to compare a RL agent with a human agent in Powderworld?


**Summary Of The Paper:**

The paper proposes a new simulation environment (Powderworld) to facilitate the study of generalization across tasks which share rules. The environment is extensible and designed for speed and modern GPU hardware. Experiments demonstrate how Powderworld can be used to study generalization in world modeling and RL.


**Summary Of The Review:**

The primary contribution is a novel tool consisting of a simulation environment. Its release might help advance the community better study an important problem in agent design. Difficult to predict the adoption of this tool but I don't see any reason to not add it to the toolbox. The experimental clarity of the 2 demonstrated uses of Powderworld (world modelinng and RL) could be improved.

-----

UPDATE: I thank the authors for their detailed feedback and engagement with the reviewers. The revised paper seems much improved. After reading the other reviews and the author response, I'm more inclined to accept this paper. It doesn't meet my bar for a strong accept though.

---

> ### Author Response · Authors · 2022-11-18
> **Response to Reviewer EMHS**
>
> Thank you for the strong recommendation! We recognize there are some unclear writing points, and **we have uploaded a revised paper** to address these points.
>
> To clarify some experimental details:
>
> > An illustration of the trajectory generation process for training and inference might help improve clarity in this part of the paper.
>
> **We have added a figure detailing the training process**. See Figure 5, or https://i.imgur.com/HnzGeb0.png
>
> Specifically, trajectories are generated by 1) generating a start state via PCG algorithm, 2) simulating the state forwards, recording every 8 timesteps, and 3) creating input-output pairs accordingly (e.g. Timestep0->Timestep8, Timestep8->Timestep16, etc)
>
> > Were each of the 5 models in Fig 5 trained on the same total amount of simulation data or not?
>
> Yes, the total amount of simulation data / training steps is equal. This is for fair comparison.
>
> >
> > Does the phrase "training data" in the line "world models trained on increasing amounts of training data display higher performance on a set of test tasks" refer to the number of tasks while keeping the total number of simulations constant across all 5 models?
>
> You are correct -- in all trials, each model is trained for 5000 iterations on a batch of 128 data points.
>
> > If yes, how exactly were these trajectories sampled?
>
> For this experiment, all trials are trained on the same *amount* of data, but we adjust the number of *initial states* that the data is generated from. For example, in the 10-task case, we train on 5000\*128 trajectories sampled from 10 unique starting states (thus the generated data may repeat). In the 1000-task case, we instead sample from 1000 possible starting states, and so on. **We have updated the paper to include more clear terminology**.
>
> > And does it mean that Fig 5 right is a sample observation (W_15) from the predicted logit probs (W').
>
> Again, your assumption is correct. Specifically we display the element that has the maximum probability (as the world model outputs logits over all elements). We have clarified this in the paper.
>
> > What exactly is a run and a trial? I found the statements "each run trains on 10x more data as the previous" and "Runs are trained for three trials and the average test loss over training time is displayed" confusing.
>
> Thank you for the feedback, I see how this terminology can be confusing. We mean to say that each parameter setting such as 10-task, 100-task ("run") is run three sepearate times ("trials"), and we show the average loss over these three trials.
>
> We have clarified the "10x more data" point, this should rather say that we increase the task number by 10x each run. Total amount of training data remains equal.
>
> > A similar question arises in Sec 5. How exactly are the 2.5M datapoints for the RL agent generated for agents restricted to T training tasks?
>
> Each time a new episode begins, the initial state is sampled from the set of T training tasks. So in the 10-task case, 2.5M datapoints are generated from 10 starting states (thus, starting states will repeat). In the 1000000-task case, each starting state is essentially unique.
>
> >*How well does powderworld support human agents? More concretely, how easy would it be to compare a RL agent with a human agent in Powderworld?
>
> It's definitely possible, since the action space of agents is similar to how a human interacts (place element at XY).
> **You can interact with a Powderworld simulation at this anonymized site:** https://resourceful-human.static.app/
>
> I attempted to solve a single sand-pushing task and got a reward of ~16.6. I basically wrote a script that pushes sand in a constant direction, which is my main strategy as a human. You can see a rollout of this strategy at: https://i.imgur.com/412AIh1.png
>
> **Here's a rough comparison of my performance compared to a PPO agent:** https://i.imgur.com/5N35t2w.png Human score is the orange line, while purple shows the mean reward of the PPO agent. If possible we will consider including human-performance levels for the various RL tasks in a camera-ready release -- this will provide good context for the strength of the agents.
>
> Given these clarified terms, along with the additional results and comparisons to the RL section, will you consider increasing your score/confidence? Please let us know any additional comparisons that would be helpful for you.

---

> > ### Comment · Reviewer_EMHS · 2022-12-06
> > **Update**
> >
> > I thank the authors for their detailed feedback and engagement with the reviewers. The revised paper seems much improved. After reading the other reviews and the author response, I'm more inclined to accept this paper and am more confident in my assessment that this would be a good addition to the toolbox. It doesn't meet my bar for a strong accept though.

---

### Official Review · Reviewer_pDhR · 2022-10-29

**Confidence:** 4
**Correctness:** 3
**Technical Novelty And Significance:** 2
**Empirical Novelty And Significance:** 3
**Recommendation:** 8

**Clarity, Quality, Novelty And Reproducibility:**

In terms of clarity and quality, it's a good paper.  I did find a few typos (apologies I did not write them down).  The environment is sufficiently novel and useful.

**Strength And Weaknesses:**

It seems difficult to find an argument against this work and this environment.  In favor of this work, the environment seems very well thoughtout, and it occupies a niche that seems to be important and unfulfilled at the moment -- an environment that is easy for users to specify novel worlds and tasks, and in which the world interactions that occur at each step are complex and take place throughout the environment (thus maybe facilitating more explorations towards more real-world scenarios, while still maintaining many of the setup and computation we enjoy with toy environments).  Frankly, if I worked in an area where this environment would be more applicable, I would be strongly inclined to use it, and therefore I'm strongly inclined to accept it.  It looks great to me!

The downsides of this work is that the experimental sections offer little in terms of useful experimental results.  The experiments exist only to sanity-check that the environment is suitable for the types of generalization experiments that are of the most interest to the authors, and to further reproduce tried-and-true beliefs about the relation between train environments, models, and generalization.  As such, assessing the merit of the paper as a publication is really (mostly) about considering the value of the software as a resource.


**Summary Of The Paper:**

The authors present a new environment for supporting RL research, especially for more physics-oriented tasks.  In the environment, each "pixel" may be occupied by a certain type of material, and the material properties and environmental forces define how the update should be updated to the next state, and can be accomplished through very localized and efficient computation.  Special care was given such that the environments are simple to specify (and have a matrix representation), a diverse number of environments is easily configurable, and with the provide set of materials and rules, model generalizability to novel situations is easy to test.  The authors apply RL and world model baselines to an example set of environment worlds.






**Summary Of The Review:**

The authors provide an environment which is:
- fast
- easily configurable
- has a small set of fixed rules and properties controlling a rich space of observable interactions
- mostly unique wrt other environments I'm familiar with

Through the paper discussion and experiments, they make a compelling case for using this environment if the task domain is relevant to ones research.  They outline a number of possible task scenarios that seem useful, but moreover, the dataset seems flexible enough to cover many unmentioned task scenarios.  I find it hard to find fault in the environment, and suggest not accepting the paper only if it is considered not publication-worthy in terms of its research content (though precedent on similar environment papers would make a good case for accepting it).

Update: I'm updating the correctness score 1 -> 3.  It was incorrectly entered and so I hope it didn't cause too much confusion juxtaposed next to my otherwise positive review.

---

> ### Author Response · Authors · 2022-11-18
> **Response to Reviewer pDhR**
>
> Thank you for your strong review.
>
> > It seems difficult to find an argument against this work and this environment. In favor of this work, the environment seems very well thoughtout, and it occupies a niche that seems to be important and unfulfilled at the moment -- an environment that is easy for users to specify novel worlds and tasks, and in which the world interactions that occur at each step are complex and take place throughout the environment (thus maybe facilitating more explorations towards more real-world scenarios, while still maintaining many of the setup and computation we enjoy with toy environments). Frankly, if I worked in an area where this environment would be more applicable, I would be strongly inclined to use it, and therefore I'm strongly inclined to accept it. It looks great to me!
>
> We appreciate the clear understanding of the purpose of this work -- to present an expressive environment for the research community that is rich yet computationally efficient. Our main focus was developing and implementing the ruleset to make this possible.
>
> > The downsides of this work is that the experimental sections offer little in terms of useful experimental results. The experiments exist only to sanity-check that the environment is suitable for the types of generalization experiments that are of the most interest to the authors, and to further reproduce tried-and-true beliefs about the relation between train environments, models, and generalization. As such, assessing the merit of the paper as a publication is really (mostly) about considering the value of the software as a resource.
>
> You are correct; the main purpose of the experimental section is to motivate the Powderworld ruleset by providing empirical evidence of generalization. We believe presenting baselines that replicate tried-and-true beliefs is crucial, as this allows researchers to be confident that Powderworld supports similar training dynamics to past environments.
>
> Specifically, experiments are presented for the following reasons:
> - **Generalization improves with more training data**. This is an expected result, presented to confirm that Powderworld-simulated data displays similar scaling properties as real-world data.
> - **Generalization improves with more complex training states**. These experiments show that the *diversity* of training states matters. These results motivate the benefit of multi-task learning, and encourage research on methods such as UED that automate task design.
> - **Models generalize to NEW elements.** We argue that this result is substantially novel and unique to Powderworld. Because of the modular nature of Powderworld, we can examine models trained on a subset of the ruleset (1 element, 2 elements, etc.). Because models trained on more elements show better transfer performance to unseen elements, it shows that models succesfully learn not only specific rules (i.e. fire burns wood), but also general properties of Powderworld *as a system* (i.e. elements are affected by gravity, elements only affect local neighbors, etc).
>
> We have additionally conducted further experiments for reinforcement learning, introducing a total of three tasks that exhibit varying generalization properties. **New results (Figures 7, 8, 9)  showcase that agents can be trained to push sand into a goal, destroy an existing scene, and direct water into a container***. These results have been added to the paper revision. Given the new results, will you consider increasing your score or correctness rating?
>
> >  I find it hard to find fault in the environment, and suggest not accepting the paper only if it is considered not publication-worthy in terms of its research content (though precedent on similar environment papers would make a good case for accepting it).
>
> We agree that there is a precedent that environment papers focus largely on baseline results, e.g. Crafter (Hafner 2021), Avalon (Albrecht 2022). We go further and provide generalization experiments similar to ProcGen (Cobbe 2020).

---

> > ### Comment · Reviewer_pDhR · 2022-12-06
> > **Response to Rebuttal**
> >
> > I'd like to thank the authors for making significant revisions to the paper in light of reviewer concerns.  As much as I appreciate these additions, I don't believe they warrant a further raise in score beyond a strong recommendation acceptance (to go further would I think require more obvious and significant impact on the field).  But it looks like a very useful contribution and I look forward to seeing what comes of it.

---

### Author Response · Authors · 2022-11-18
**Rebuttal Revision**

To all reviewers -- thank you for the helpful feedback! We received largely strong recommendations, along with suggestions for improving the experimental section.

We have submitted a revised paper with a thoroughly updated reinforcement learning section. Specifically, changes are as follows:
- Simulation speed comparisons are presented to related RL environments (Fig 2)
- Clarifications to the training procedure for world model experiments (Fig 5)
- **We extend the sand-pushing task into a general framework for RL tasks in Powderworld (Fig 7, 9)**. Within this framework, we present three domains: 1) sand-pushing; where an agent creates wind to push sand into a goal, 2) destroying; where an agent must place a small # of elements to efficiently destroy the world, and 3) path-building; where an agent builds/removes walls to direct water into a container.
- For each RL domain above, we conduct experiments where the complexity of agent training tasks are increased (Fig 8). Each trial is run with 5 random seeds. **We show that increasing complexity can help generalization in RL, up to a task-specific inflection point**. To achieve these results (vs. previous results), it was important to switch to a discrete action space and increase PPO batch size.
- A anonymized website is presented with an interactable Powderworld simulation and videos of the RL tasks: https://resourceful-human.static.app/

Reviewer feedback generally supported the Powderworld environment, and requested additional RL experiments. We hope these updates satisfy this request. **If so, we ask that you consider raising your score, and please let us know any changes that would be helpful for a potential camera-ready version.**

As the presented environment remains the same, we believe these updates are not a substantial deviation from the original work, but rather present a more thorough exploration.

---

### Decision · Program_Chairs · 2023-01-20

**Decision:**

Accept: notable-top-25%

**Justification For Why Not Higher Score:**

I am torn on this one between Spotlight and Oral. The rationale for pushing for a higher score is that the final paper after the rebuttal period is exemplary for a paper proposing a new benchmark, and for an empirical paper more generally. The paper is systematic, claims are well supported, experiments serve a clear purpose and are well designed and thorough. Highlighting this more can set a good example to the research community.

**Justification For Why Not Lower Score:**

The rationale for a lower score is the breadth of audience this paper will likely impact. The proposed benchmark focuses on generalization in reinforcement learning and model learning. These are important, but only relevant to a subset of ICLR attendees.

**Metareview: Summary, Strengths And Weaknesses:**

The paper introduces a new research environment and benchmark tasks, Powderworld, designed to drive research on generalization in learning world models and in reinforcement learning. The environment is systematically developed to address key challenges, and the authors demonstrate the type and depth of insight that can be developed using this environment by studying questions of generalization in world model learning and reinforcement learning.

Reviewers initially raised some concerns about this work, including that initial experiment did not fully demonstrate the value of the platform, need for more clarity in describing experiments, sources of bias, and clarity on what kinds of research the platform would support.

The authors provided detailed responses to reviewers, and provided a new version of the paper that addressed reviewer concerns. All reviewers agree that the resulting version of the paper is stronger than the initial submission, and that it addresses the concerns reviewers raised initially.

In its present form, the paper makes a valuable novel contribution, is well written, clear, and systematic. The consensus is to recommend acceptance.

**Note From Pc:**

if the above contains the word "oral" or "spotlight" please see: "oral" presentation means -> notable-top-5% and "spotlight" means -> notable-top-25%. As stated in our emails, we are disassociating presentation type from AC recommendations